# Variation in natural infection outcomes and cancer cell release from soft-shell clams (*Mya arenaria*) with bivalve transmissible neoplasia

Rachael M. Giersch[1,2☉], Jordana K. Sevigny[1,3☉], Sydney A. Weinandt[1,4☉], Carissa Mayo[1,5], Fiona E. S. Garrett[1], Karyn Tindbaek[1,6], Marisa A. Yonemitsu[1,7], Samuel F. M. Hart[1,7,8], Michael J. Metzger[1,5,7]*

1 Pacific Northwest Research Institute, Seattle, Washington, United States of America, 2 Institute of Molecular Biology, University of Oregon, Eugene, Oregon, United States of America, 3 Department of Ocean Sciences, University of California Santa Cruz, Santa Cruz, California, United States of America, 4 Department of Biology, University of Alabama at Birmingham, Birmingham, Alabama, United States of America, 5 Quantitative Ecology and Resource Management Program, University of Washington, Seattle, Washington, United States of America, 6 Fred Hutchinson Cancer Center, Seattle, Washington, United States of America, 7 Molecular and Cellular Biology Program, University of Washington, Seattle, Washington, United States of America, 8 Department of Genome Sciences, University of Washington, Seattle, Washington, United States of America

☉ These authors contributed equally to this work.

* metzgerm@pnri.org

## Abstract

Bivalve transmissible neoplasias (BTNs) are leukemia-like cancers found in at least 10 bivalve species, in which the cancer cells themselves transfer from one individual to another, spreading as an unusual form of infectious disease. Before the infectious etiology was known, there were reports of lethality and outbreaks of cancer in the soft-shell clam (*Mya arenaria*) on the east coast of North America. Using sensitive and specific qPCR assays, we followed the outcomes of BTN in naturally-infected soft-shell clams from Maine, USA. We observed variable outcomes, with about half of clams (9/21) progressing to high levels of cancer and death, about half exhibiting long-term non-progression (11/21), and a single animal showing regression of cancer. We also observe a significant decrease in survival in animals that progress to >10% cancer in their hemolymph, while we see no effect on survival in clams with BTN that are long-term non-progressors. As most bivalves do not physically contact each other, and BTN cells can survive in seawater, it has been proposed that BTN is spread through release of cancer cells into the water. We used qPCR to detect BTN-specific sequences in environmental DNA (eDNA) in the tanks of animals throughout this experiment. We show that BTN-specific eDNA (likely from released cancer cells) can be detected in tank water of most clams with >24% cancer in their hemolymph, but not below this level. This detection of BTN eDNA is variable and occurs in bursts, but in clams with >24% cancer, the detection of BTN eDNA correlates with progression of the cancer in the hemolymph.

**Data availability statement:** All raw data on clams, their survival, and all qPCR results are provided in Supporting Information.

**Funding:** This work was funded by a grant from the National Cancer Institute of the National Institutes of Health, USA, nih.gov (R01CA255712) to support work on disease progression and cancer resistance (MJM) and an Ecology and Evolution of Infectious Disease grant from the National Science Foundation, USA, nsf.gov (2208081) to support eDNA analysis and disease transmission (MJM). SFMH was supported by NIH training grants T32-HG000035 and T32-GM007270. The funders had no role in study design, data collection and analysis, decision to publish, or preparation of the manuscript. The content is solely the responsibility of the authors and does not necessarily represent the official views of the National Institutes of Health.

**Competing interests:** The authors have declared that no competing interests exist.

This study demonstrates the lethality of BTN, but the observation that about half of clams with BTN do not progress to death provides evidence suggesting that there may be a block to the progression of BTN in a large portion of clams in a population with this enzootic disease. This study also further supports the hypothesis that BTN cells transmit through seawater and provides insights into the mechanisms of the transmission dynamics.

## Author summary

Bivalve transmissible neoplasia is an infectious disease in which cancer cells themselves are the infectious agent, jumping from clam to clam through the environment. We maintained soft-shell clams from Maine with low levels of bivalve transmissible neoplasia in the lab, quantifying the amount of the cancer cells in their hemolymph and the amount released into the water of their tanks. We found that about half of the clams progressed to high levels of cancer and death, while half maintain the cancer at a low level without progression, and a few show evidence of cancer increase followed by regression. Study of tank water led to our observation of two stages of disease: an early stage (below 24% cancer in the hemolymph) in which very little cancer-specific DNA was observed in the water, and an infectious stage at higher levels of cancer in which high, but variable, levels of cancer-specific DNA were observed. Overall, this study defines the variable outcomes of transmissible cancer, identifies a long latent period, characterizes the stages at which cancer cells are released, and suggests that some clams have evolved an as-yet-unknown mechanism for preventing transmissible cancer growth or engraftment.

## Introduction

Bivalve transmissible neoplasia (BTN) is a naturally occurring transmissible cancer affecting multiple bivalve species around the world. While most cancers develop from oncogenic mutations and remain within the same individual, there are three types of transmissible cancers that have been observed in nature and act as a previously unrecognized mode of infectious disease [1]. These include one lineage of canine transmissible venereal tumor (CTVT) in dogs [2,3], two lineages of devil facial tumor disease (DFTD) in Tasmanian devils [4,5], and at least 10 lineages of BTN across 10 different bivalve species [6–13]. Each de novo lineage of a transmissible cancer originates from an individual founder animal from its own respective species and subsequently spreads to other individuals, outliving the original host.

While CTVT and DFTD are solid tumors, BTN is a leukemia-like neoplasia that has been described as disseminated neoplasia (DN) or hemic neoplasia, before the infectious etiology was known. DN was first reported in the 1960s, it is characterized by rounded polyploid cells that are observed in high numbers in the

hemolymph of the bivalve that disseminate into the tissues of the animal at later stages of disease, and it has been observed in more than 20 species of bivalves [14–16]. Outbreaks of DN in soft-shell clams (*Mya arenaria*) led to greater than 90% prevalence on the East Coast of the United States in the 1980s and in Prince Edward Island, Canada, in the 2000s. These outbreaks were associated with up to 90% population loss in these areas [17,18], suggesting that the disease is, or at least was, highly lethal. No notable DN outbreaks associated with die-offs have been reported in New England since then, but the disease is routinely observed at low levels [8,19,20], and is considered enzootic in these populations.

Most studies of DN have been done with single, lethal collection and diagnosis, which does not allow for understanding of the long-term outcomes of the disease. Progression dynamics of DN have been studied in a few instances including both laboratory and field studies. Cooper et al. diagnosed clams and observed several different outcomes: progression to death at high levels of infection, maintenance of the disease at lower infection levels, and remission from lower levels [21]. In this experiment, animals were continuously exposed to each other in flow-through tanks and cancer levels were determined at the start of the experiment using histological analysis of hemolymph. Another study reported that, in both field and laboratory settings, high severity neoplastic animals died at a higher rate than low severity neoplastic animals [22]. While keeping the animals in flow-through tanks and using histological analysis of hemolymph, these studies attempted to observe the length of progression and mortality rate. These progression dynamics studies combined with population loss in infected areas subsequently lead to reports that DN is a lethal disease with some cases of remission and maintenance of the disease. While these studies were done before the etiology as a transmissible cancer was known, it has been estimated that the BTN that is observed currently in *M. arenaria* (MarBTN) is more than 200 years old [23], so it is likely that the DN observed in these studies was transmissible cancer from that same lineage, although this cannot be confirmed.

Both CTVT and DFTD are transmitted via direct physical contact between individuals. In contrast, *Mya arenaria* and other bivalves are sessile, making direct interaction an unlikely route for BTN transmission. Instead, BTN most likely spreads through the water column. This transmission mechanism requires several steps: the release of cancer cells from diseased clams, survival of the cells in seawater, and engraftment into naïve clams. Bivalves obtain nutrients by filtering large amounts of seawater, during which it is possible that neoplastic cells are drawn up or released by the bivalve, although the exact mechanism is unknown. Previous studies have shown that BTN cells from *M. arenaria* and other bivalves can survive for days to months in seawater environments [19,24,25], and it has been shown that MarBTN-specific DNA could be observed in tank water of a heavily diseased clam [19]. The presence of MarBTN-specific DNA in wild seawater samples has also been detected in environmental DNA (eDNA) near infected *M. arenaria* populations in Puget Sound, Washington, USA [26]. Together, these two lines of evidence (1, *ex vivo* BTN cell survival in a range of biologically relevant environmental conditions and 2, detection of neoplastic DNA in the seawater around wild MarBTN positive populations) support the hypothesis that BTN cells are transmitted through the water column. However, it is not known when or for how long a clam is infectious, and how much or how often diseased clams release BTN cells into their environment.

We have shown that the DN that has been observed in soft-shell clams (*M. arenaria*) is a bivalve transmissible neoplasia, and we have developed sensitive allele-specific qPCR primers to determine the severity of the neoplasia [8,19]. These new methods enable an analysis of BTN disease dynamics that is more sensitive and accurate than previously possible. Notably, while DN is largely considered to be a fatal disease, we have anecdotally observed cases of apparent regression of BTN during maintenance of diseased clams in the lab. We therefore designed this study to use molecular markers to determine the timing and outcomes of BTN in clams, using analysis of the fraction of MarBTN cells in the hemolymph to quantify cancer progression, and collecting and analyzing eDNA from tank water to quantify the release of BTN DNA throughout the course of the disease. These analyses greatly inform our understanding of the impact of BTN on bivalve populations and the mechanisms of disease transmission.

## Methods

### Collection of clams and initial hemolymph sampling

Adult soft-shell clams (*M. arenaria*) >50 mm in length were collected by commercial sources from multiple locations in Maine, USA (S1 Table). In Maine, 2 inches is the minimum size limit for collection. After shipping of clams on ice to the PNRI lab in Seattle, WA, approximately 0.5–1 mL of hemolymph was collected from the pericardial sinus of each animal using a 0.5 inch 26-gauge needle fitted on a 3 mL syringe and stored in a 1.7 mL Eppendorf tube kept on ice for the duration of the sampling. Approximately 20–50 µL of hemolymph was also placed in a 96-well plate and incubated at 4°C for 1 hour to allow the cells to settle. Imaging of the hemolymph cells was performed on an inverted phase-contrast microscope to visualize the healthy and neoplastic cells, as done previously [8,19]. The remaining volume of hemolymph was spun down in a centrifuge at 1,000 × g, 4°C to pellet the cells. The supernatant was removed, and the cells were stored at -80°C until DNA extraction. DNA extraction of hemocytes was performed using DNeasy Blood & Tissue Kit (Qiagen, Hilden, Germany).

### Identification and experimental setup of low-positive and negative clams

Animals were diagnosed upon arrival in the lab using a sensitive qPCR assay. As our study was designed to follow animals throughout the course of infection, we arbitrarily selected 10% MarBTN cells in the hemolymph as the cutoff for inclusion in this study. Clams with low cancer levels (defined as detectable, but <10% MarBTN cells in the hemolymph) were identified by qPCR, and a set of negative, size-matched clams from the same location and collection as the low-positive clams were identified (S1 Table). These experimental animals were housed for at least two weeks prior to initiation of the experiment to minimize risk of false negative diagnosis, to allow adjustment to the lab environment, and because clams that are damaged during shipping often die within this period.

For the duration of the experiment, animals were individually housed in 1.8 L of 1 × Artificial Sea Water (ASW) (33 g/L Crystal Reef Salt, Blacksburg, VA, USA) with constant aeration at 16°C. The 1 × ASW was replaced for each clam twice per week (replacing between 50% to 100% of ASW), and after water changes the animals were fed twice per week with PhytoFeast or LPB Frozen Shellfish Diet (Reed Mariculture, Campbell, CA, USA).

Hemolymph was sampled every two weeks throughout longitudinal follow-up of the clams (47 initially diagnosed as low-positive and 47 as negative controls), with microscopy images taken and DNA extracted from hemocytes, as described above.

After data collection, clams were included or excluded from analysis based on the following criteria. Initially identified low-positive clams that had > 10% BTN cells in the hemolymph at Day 0 were excluded, and initially identified negative clams that had detectable cancer at Day 0 were excluded. In order to remove false-positive clams, low-positive animals were excluded from analysis if there were no timepoints beyond the initial screening at which the hemolymph was positive by qPCR for MarBTN (in all three replicate qPCR wells). Animals were also excluded from analysis if they died within 8 weeks of the start of the experiment (Day 0). After excluding these animals, 21 natural low-positive animals and 39 negative control animals remained.

### Aquaria eDNA collection and extraction

Water samples were collected for eDNA analysis from all tanks once per month, or once every two weeks following a diagnosis of >10% MarBTN. For each water collection, ASW in the tank was completely replaced 24 hrs before collection, and 250 mL of ASW was collected immediately after stirring the tank, just prior to hemolymph collection. Water samples were vacuum-filtered through a 47 mm diameter, 0.45 µm cellulose nitrate filter. Using clean forceps, the sample filter was folded small enough to fit into a 2 mL tube and frozen at -80°C until eDNA extraction. The eDNA extraction protocol was modified from Renshaw et al. [27]. Briefly, 900 µL CTAB buffer (2% CTAB *w/v*, 20 mM EDTA, 100 mM Tris-HCl, and 1.4

M NaCl, in water) was added to the filter, and the tubes were incubated at 65°C for 1 hr with shaking at 300 rpm (water samples from the tanks of animals FFM-18B12, FFM-16E1, and FFM-15A12 were extracted using an earlier version of the protocol, incubating in CTAB for 30 min without shaking, which may have resulted in a slightly lower eDNA yield for these samples). Tubes were spun to collect the sample in the bottom of the tube and 900 µL chloroform:isoamyl alcohol (24:1) was added, followed by shaking or vortexing. Samples were spun for 5 min at 15,000 × g, and the 700–850 µL aqueous layer was transferred to a new tube with 700 µL chloroform. This was shaken and spun as before, and the ~700 µL aqueous layer was transferred to a new tube containing 700 µL cold isopropanol and 24 µL 5M NaCl. 4.67 µL glycogen blue was added to ensure visibility of the pellet, and samples were allowed to precipitate overnight at -20 to -30°C. DNA was spun for 10 min at 15,000 × g and the liquid removed by pipette. 500 µL of 75% ethanol was slowly added, the tube was spun a second time, and the ethanol was poured off. DNA pellets were air dried and resuspended in 100 µL Buffer EB (Qiagen).

## qPCR of DNA from hemolymph

To quantify the amount of neoplastic and host cells in a hemolymph sample, an allele-specific qPCR was performed using two pairs of primers. The MarBTN-specific primer pair (ClamLTR-F3: TTCAATCATTCAACGCATAACC and N1N2can-R3: TCGCTGAGAATTTTTCGGTGT) amplifies a single integration site of the LTR-retrotransposon *Steamer*, near the N1N2 ORF [19]. *Steamer* is present in only a few copies in healthy clam genomes but highly amplified in MarBTN [23,28], making any *Steamer* insertion highly likely to be a somatic mutation that is specific for MarBTN cells. Moreover, this specific insertion has been observed in all samples of BTN in *M. arenaria* to date and has not been detected in genomic DNA of any healthy clam, further justifying its use as a specific marker of MarBTN. The MarBTN-specific primer pair targeting this insertion junction amplifies half the total amount of N1N2 alleles in a cancer cell, as the insertion is present in two of four copies of the gene in a region that is tetraploid in known samples of MarBTN from USA [19,23]. The host-specific "universal" primer pair (N1N2-F3: CCCAGGGCAAGAGGAATATGGT and N1N2-R1: GGATACTGCAAGCTTCTTGGAA) targets a conserved region of the N1N2 ORF nearby and quantifies the total copies of the N1N2 locus present. As the host cells are diploid and cancer cells are polyploid, the fraction of cancer cells in the hemolymph sample is calculated by the equation:

$$\frac{P_N R}{Q_C + (P_N - P_C)R}$$

Where $P_N$ is ploidy of normal cells, R is the ratio of cancer alleles to total alleles measured by qPCR, $Q_C$ is the number of "cancer alleles" in cancer cells, and $P_C$ is the ploidy of cancer cells (derivation of this equation in S1 Text). For samples from the USA, in which 2/4 copies of the locus contain the cancer target, the fraction of MarBTN cells in the clam hemolymph can be reduced to the equation: R/(1-R). A single plasmid (pCR-SteamerLTR-N1N2) was used for the standard curve for absolute quantification of DNA copy number for both primer pairs. The plasmid concentration was measured (Qubit, Thermo Fisher Scientific) and copy number per µL was calculated based on the plasmid size. Plasmids were linearized with 0.25 µL of NotI-HF (NEB, Ipswitch, MA, USA) for 30 min at 37°C in a 20 µL reaction at $1 \times 10^{10}$ copies/µL, heat-inactivated 20 min at 65°C, then diluted to $1 \times 10^9$ with 180 µL Buffer AE (Qiagen). Standard curves were prepared from $1 \times 10^7$ copies/rxn to $1 \times 10^1$ copies/rxn. For all DNA samples, 2 µL of extracted DNA was run in 10 µL reactions on a StepOnePlus real-time PCR cycler (Applied Biosystems, Waltham, MA, USA). Reactions were run as follows: 95°C for 2 min, 40 cycles of 95°C for 15 s and 60°C for 30 s, followed by a melt curve using 95°C for 15 s, 60°C for 1 min, and ramping 0.3°C from 60°C to 95°C, followed by a 15 s hold at 95°C. All samples were run in triplicate and values presented are an average of triplicates. When averaging triplicate values, undetectable amplification was treated as zero, and any average value below 1 copy/rxn (the limit of detection) was treated as 1 copy/rxn.

### Survival analysis

To quantify the impact of BTN on survival, we conducted a Kaplan-Meier analysis using the survival (version 3.5-5) and survminer (version 0.4.9) packages in R (Therneau, T. M. 2023; Kassambara, A., & Kosinski, M. 2021)[29,30]. First, we conducted a Kaplan-Meier analysis of survival using 1-to-1 matching of low-positive clams with control clams, size matched and from the same collection (21 pairs). Second, we conducted a separate Kaplan-Meier analysis focused on low-positive animals that reached at least 10% neoplasia (n = 9), along with size-matched healthy controls from the same collections (n = 9). For this subanalysis, the survival time began the first day a low-positive animal was measured at >10% neoplasia, with their size-matched control animal's survival time beginning at the same point, through the death of the final animal.

### qPCR of MarBTN in eDNA from tank water, changepoint analysis, and linear regression

Quantification of MarBTN-specific eDNA from tank water was done using the qPCR assay described above. The absolute copy number per reaction of MarBTN-specific sequences in eDNA (the average of triplicate reactions) was first transformed by taking the neoplastic DNA copies per reaction/ 2 µl of eDNA per reaction × 100 µl total extracted eDNA/ 250 mL (the volume collected from each tank) to obtain copies/mL, wherein the limit of detection (1 copy/rxn) was 0.2 copies/mL.

To determine the relationship between disease progression and MarBTN cell release, we used all collection points from all clams included in the study where hemolymph and eDNA were both available. We transformed the data into Bernoulli format, where a value of 1 indicated detectable neoplasia, (above the limit of detection, 1 copy/rxn), and 0 indicated undetectable neoplasia in the water. Then we used the changepoint package in R [Killick, 2014] [31], applying the BinSeg (binary segmentation) method and Q = 1 to identify the single most likely point at which clams began emitting neoplasia particles based on the observed data and to reduce risk of overfitting. After determining the changepoint, we performed a linear regression on the log-transformed MarBTN values of eDNA collected from clams with cancer above the changepoint.

## Results

### Anecdotal case of regression of MarBTN

Prior to this study, we had maintained soft-shell clams (*M. arenaria*) with MarBTN in aquaria in the lab and had collected hemolymph samples for multiple experiments, including developing the quantitative assays and testing cancer cell survival [19]. In one case, we had collected hemolymph multiple times from a clam with low levels of disease, but instead of progressing to higher levels and to death, as expected, the cancer instead appeared to regress (Fig 1). This anecdotal case of regression led to the development of a more careful and controlled analysis of the progression of natural infection of clams with MarBTN presented in this study.

### Variation in disease outcomes

To determine the natural course of MarBTN in naturally infected animals and whether progression to death or regression of the cancer were more common outcomes, we tested hemolymph from clams (*M. arenaria*) that were collected from wild populations in Maine using a sensitive qPCR assay and identified animals with detectable but low levels of MarBTN (<10% cancer cells in hemolymph). We then maintained these animals and control animals (negative for MarBTN) in isolated tanks and followed disease outcomes over time with qPCR of hemolymph samples every 2 weeks until the death of the animals. Quantification of the fraction of MarBTN cells in the hemolymph using qPCR agreed with microscopic analysis of those same samples (S1 Fig). The analysis of the fraction of MarBTN cells in the hemolymph shows that naturally infected animals have significant variation in disease dynamics and outcomes (Fig 2). Based on the data we observed, we categorized animals into three main outcome schemes: progression to death (surpassing and maintaining >10% of

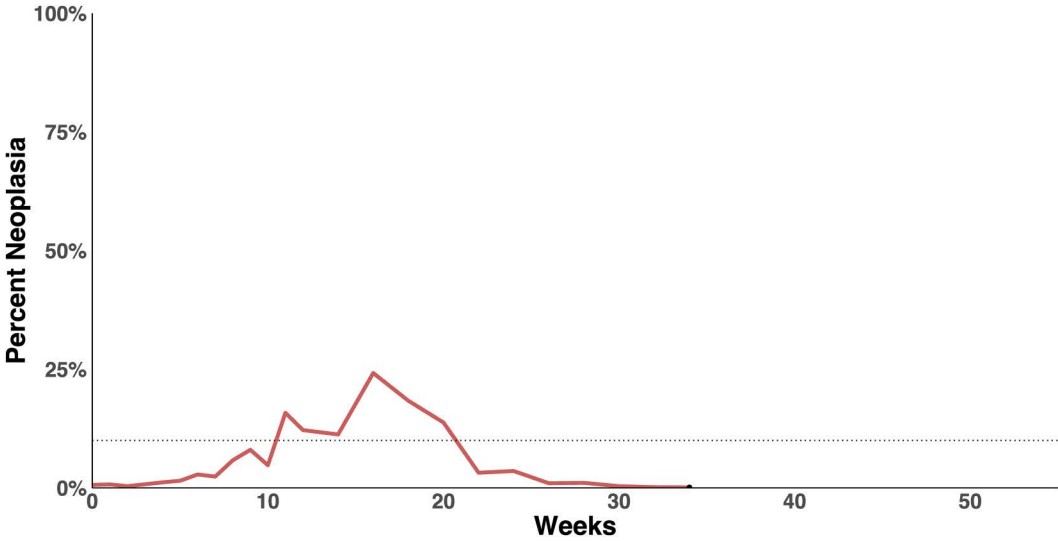

**Fig 1. Anecdotal case of regression in a clam with naturally acquired BTN.** While maintaining soft-shell clams (*Mya arenaria*) in laboratory conditions, we repeatedly sampled hemolymph and used a sensitive qPCR assay to determine the fraction of the cells in the hemolymph that are MarBTN. In the case of one clam (MLN-4F2, collected from the wild by a commercial source in Maine), we noticed that instead of progression to death, the clam appeared to progress to >20% MarBTN, but then the cancer regressed, and the animal continued to survive for several months. Black dot marks final hemolymph sample before death.

neoplastic cells in hemolymph), long-term non-progression (maintaining <10% of neoplastic cells in hemolymph throughout the experiment), and regression (surpassing 10% followed by at least a 50% decrease in fraction to below 10% for at least three time points). We found that approximately half of the clams progressed until death (9 of 21: 42.9%), and there was only a single clear case of regression of cancer (1 of 21: 4.8%). Unexpectedly, just over half of the clams confirmed by multiple qPCR samples to have low levels of MarBTN maintained low (<10%) or nearly undetectable levels of disease (11 of 21: 52.4%). This confirms that MarBTN regression can occur, but it is rare, and the more common outcomes are progression to death or long-term non-progression.

Of the 39 negative control animals, we observed two cases of progression and one case of regression, despite the fact that all initially were diagnosed as qPCR-negative at the experiment start date (Fig 2). Progression of MarBTN in these three animals shows that there is sometimes a significant latent period in natural disease, in which the clam has been exposed and harbors MarBTN cells somewhere in the body, but MarBTN cells are undetectable in the hemolymph.

## Neoplastic animals have decreased survival

We first quantified mortality rates of the negative control clams and naturally infected neoplastic clams to determine if infection correlates with decreased survival. When comparing survival of all clams that started with low levels of MarBTN to those that started negative for disease, we observed a trend towards lower survival with MarBTN, but when we corrected for bias in collection site by matching cases and controls, the difference is not significant (Figs 3A and S2). There are several possible reasons for this, including the presence of latently diseased animals in the control group in which MarBTN was undetectable at the start of the experiment, and the major finding that the cancer does not progress in the majority of clams. In order to test more specifically whether progression of MarBTN was associated with mortality, we conducted a subanalysis of survival after the date at which a progressing clam reached 10% cancer in their hemolymph (Fig 3B), compared to survival of a matched negative clam starting at that same timepoint. In this analysis, it is clear that progression to >10% of MarBTN in the hemolymph leads to decreased survival (p = 0.022; Fig 3C). While low-level cancer

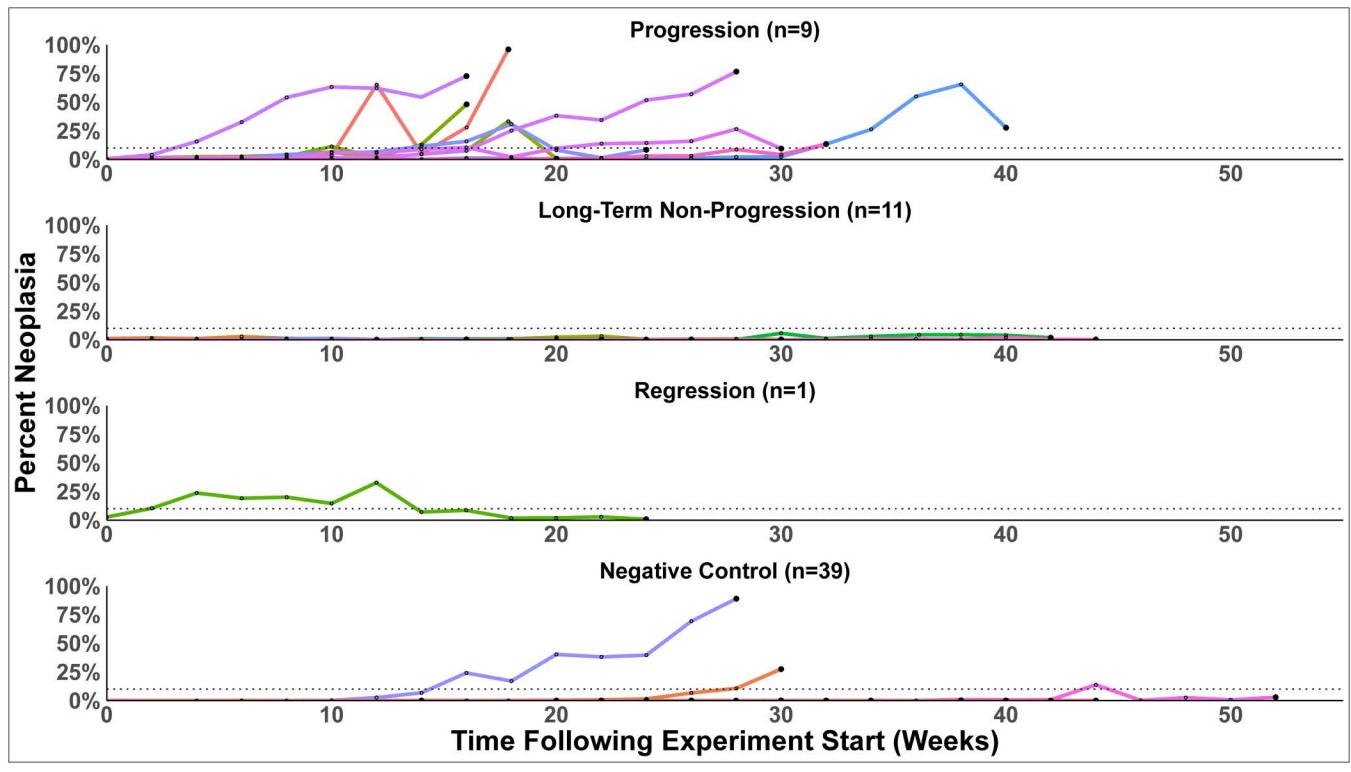

**Fig 2. Multiple distinct outcomes observed in soft-shell clams (*Mya arenaria*) after natural infection with MarBTN.** Soft-shell clams were collected from multiple sites in Maine, USA, diagnosed for MarBTN using qPCR, and we selected 21 low-positive clams (<10% MarBTN in hemolymph) and 39 negative control clams (undetectable for MarBTN-specific DNA) to follow over time, sampling hemolymph every two weeks until death. Each colored line is a separate clam. Large dots represent final hemolymph draw before clam death. Dotted line shows 10% cancer. Animals that started with low cancer have been split into three categories reflecting different outcomes: Progression (n = 9), Long-term non-progression (n = 11), and Regression (n = 1). Negative control clams (n = 39) are all shown on the bottom.

appears not to have impacted survival of the clams that were long-term non-progressors compared to their healthy counterparts, those that did progress did have a significantly lower survival rate, further supporting the historically observed lethality of the disease.

## Detection of MarBTN-specific eDNA in tank-water of infected clams correlates with progression of disease

In addition to detection of MarBTN cells in the hemolymph, we also collected water samples throughout the course of disease, extracting eDNA and testing for the release of MarBTN-specific eDNA that likely reflects release of the cancer cells. Overall, healthy control animals displayed essentially no evidence of the disease in the water column, as expected, while MarBTN-specific DNA could be readily detected in many samples of eDNA extracted from tank water of diseased clams (Fig 4). Notably, no neoplastic DNA, or very low levels, was detected throughout months of sampling of long-term non-progressor clams, but the clams that progressed to higher levels of MarBTN showed release of high levels of MarBTN eDNA. Additionally, in the clam in which the cancer regressed, neoplastic eDNA was only found in the water at the same time as the highest amount of cancer in its hemolymph and was not detectable before or after, again supporting the idea that cancer cell release appears to be restricted to high levels of disease, and showing that it can be stopped if the disease stops.

High variation in detection of MarBTN in eDNA was observed between samples, even in clams with high amounts of cancer, but the lack of detection at lower levels of the cancer suggests that there are two stages of disease: a stage in

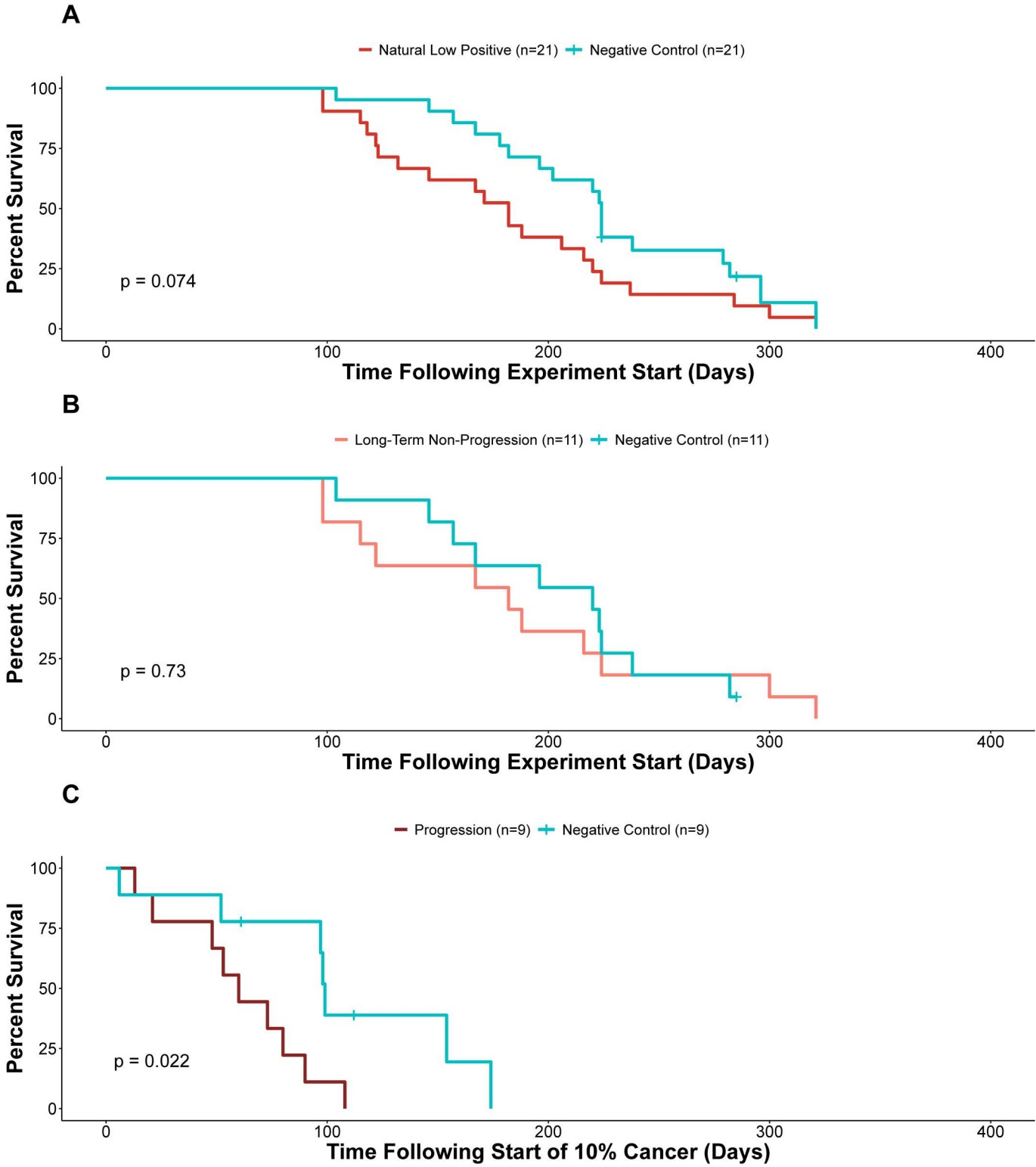

**Fig 3. Survival of soft-shell clams with MarBTN compared to control clams.** (*A*) A Kaplan-Meier survival curve compares naturally-infected clams with low levels of MarBTN (<10% at Day 0, n = 21, red line) with paired negative controls (MarBTN undetectable at Day 0, blue line). (*B*) Analysis of long-term non-progressing clams (n = 11, light red line), along with their respective paired controls (blue line), shows no difference in survival with maintained

low-level infection. (C) A sub-analysis of low-positive clams in which MarBTN progressed and did not regress (n = 9, dark red line), with survival starting at the time the clam was detected with >10% cancer, compared with their paired control clams starting at the same date (blue line), shows a significant decrease in time-to death. "+" marks the dates at which two negative control animals were culled before natural death (these events were censored in the survival analysis).

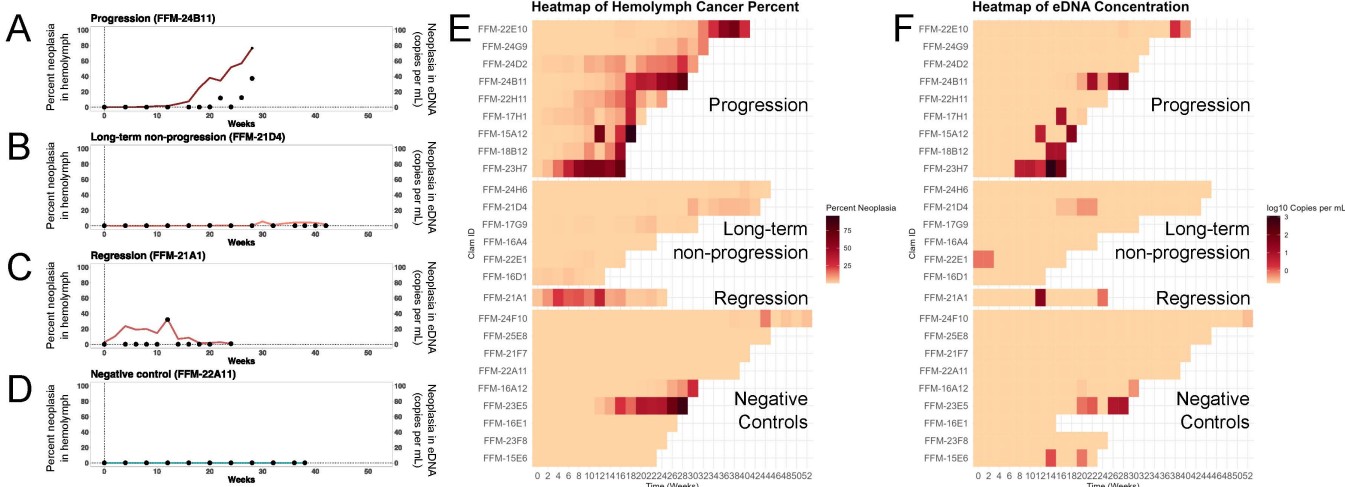

**Fig 4. Detection of MarBTN-specific eDNA in tank water at late stages of MarBTN progression.** For all cases of progression (n = 9) and regression (n = 1), and a subset of cases of long-tern non-progression (n = 6) and negative controls (n = 9), eDNA was extracted from tank water collected throughout the study, and MarBTN-specific eDNA was detected using qPCR. Dual axis plots compare the amount of MarBTN cells in the hemolymph (red and blue lines, left axis) with quantity of MarBTN-specific eDNA per mL in the tank at the time of collection (black points, right axis). Representative clams are shown for (A) progression, (B) long-term non-progression, (C) regression, and (D) negative controls (see S1 Fig for comparison with microscopy and S2 Fig for plots of qPCR measurements for all individuals). A heatmap (E) of the amount of MarBTN in hemolymph over time for multiple clams of each category is shown next to a parallel heatmap (F) of detection of MarBTN-specific eDNA showing that within the progression of each animal, MarBTN DNA release occurs mostly at later stages of disease progression.

which clams are exposed but non-infectious, followed by an infectious stage in which MarBTN cells are released. By directly comparing neoplastic DNA levels in hemolymph and eDNA across all timepoints where both sample types were available (at least once per month), we performed a changepoint analysis that identified a threshold of approximately 24%, above which the neoplastic eDNA in the water became more likely to be detected. Below this threshold, clams were largely non-infectious (Figs 5 and S3). Of the hemolymph samples that were below this changepoint (24% neoplasia in the hemolymph), only 7% of the corresponding water samples had detectable MarBTN eDNA (n = 13/203 sample times). After progression of MarBTN to higher levels in the hemolymph, 75% of timepoints were positive for MarBTN eDNA (n = 21/28). Of the clams for which we observe multiple weeks above 10% cancer, we can observe clams alternating between having high levels of neoplastic DNA in the water column and no detectable neoplastic DNA in the water column, supporting the idea that the cancer is released in bursts (Fig 5). A linear regression of the log-transformed values above this change-point showed that there is significant positive relationship between the fraction of neoplasia in the hemolymph of infected clams and the copies of neoplastic eDNA in the water column (p < 0.005). This shows that, even though there is significant variability, which may come from bursts of release of MarBTN cells, clams release more of the cancer cells as the disease progresses. Taken together, these data show that late-stage neoplastic animals release transmissible MarBTN cells and that low-level infections are unlikely to contribute to the spread of the disease and further support the hypothesis that the water column is the route of disease transmission of BTNs.

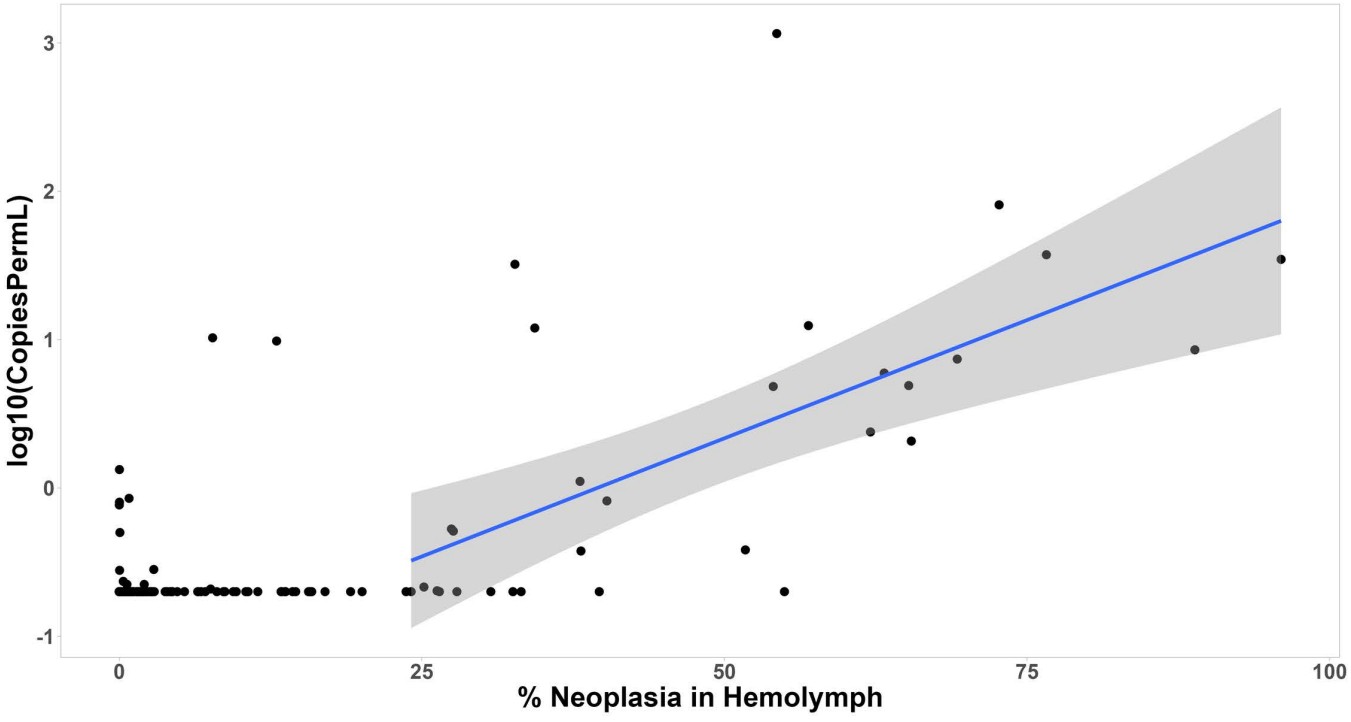

**Fig 5. Modeling the relationship between cancer level in hemolymph and MarBTN in tank eDNA.** For each timepoint for each clam at which both eDNA and hemolymph qPCR results are available, we plot a point comparing the two. Level of MarBTN in the clam is determined by the percent of MarBTN cells in hemolymph based on qPCR analysis of hemocyte genomic DNA, and cancer cell release within the 24 hrs before collection is estimated with the log transformed quantity of MarBTN-specific eDNA extracted from tank water. The vast majority of samples from clams with undetectable and very low levels of cancer in the hemolymph are associated with no MarBTN DNA release into the tank water, although there a small number of cases with detectible levels. Changepoint analysis (S2 Fig), shows that MarBTN-specific DNA is rarely detected in tank water when clams have less than 24% MarBTN in their hemolymph, so a linear regression was fit to the data points above this value (blue line, grey area marks 95% confidence interval). This shows a significant positive correlation between MarBTN-specific DNA in the water and disease progression at MarBTN levels above 24%. For the linear regression, there was an intercept of -1.2586, coefficient of 3.1869, and p-value of 0.0001258.

## Discussion

Our longitudinal analysis of soft-shell clams naturally infected with MarBTN has identified multiple outcomes to this cancer infection. We confirm MarBTN regression can be observed in clams, but this remains a rare outcome, while most MarBTN infections lead to progression and death or long-term non-progression. It is difficult to directly compare due to the different methods used before the etiology of the disease was known and due to the higher reported MarBTN prevalence of disease during the time of those previous studies, but our results are generally in agreement with previous studies of DN outcomes [21,22]. It is notable that, while a rare outcome, regression of this cancer was observed both in the current study and in the Cooper et al study, confirming that this is a consistently observed phenomenon in this disease.

There is currently no known mechanism to explain why cancer regresses or does not progress in approximately half of the infected clams and why the other half will progress to death. Susceptibility could be, in part, due to environmental exposures or other stresses that the clams had been exposed to, but the controlled nature of the longitudinal study means that during the course of observation the progressors and non-progressors were exposed to the same conditions. Clams, like other bivalves and invertebrates, have no adaptive immune system or known MHC self/nonself recognition. It seems likely then that a significant portion of the variation in susceptibility is due to genetic factors. Clam populations in Maine and throughout the east coast of North America have been exposed to MarBTN, at least since DN

was first reported in the 1960s, and outbreaks correlated with severe population loss were observed in New England in the 1980s. These early reports of widespread population loss, along with reports at the time showing that DN could be lethal in clams, and the recent observation that the lineage is more than 200 years old all combine to support the idea that MarBTN has been a significant selective pressure in clam populations for many generations. It is plausible that this selective pressure could have led to the evolution of a block to this transmissible cancer in clam populations such as the ones in New England, in which MarBTN is enzootic. Alternatively, it is possible that the MarBTN cells themselves have evolved to decrease pathogenicity (as in the case of some pathogens that can increase their spread by decreasing their pathogenic effect on their hosts), but since we observe here that infection with MarBTN without progression to high levels of disease does not appear to lead to release of cancer cells, it is unlikely that this would be a strategy that could lead to greater transmission of the cancer cells. Evolution of some unknown mechanism to prevent cancer cell growth or engraftment therefore currently appears to be the most likely cause of the decreased prevalence of MarBTN over time in New England.

Additionally, this study found that approximately half of the low-positive clams were resistant to MarBTN progression, but the total resistant fraction of the clam population may be higher, as the study only followed animals identified with detectable MarBTN when they were collected from the wild. It is possible that there is a fraction of the population that is completely resistant to MarBTN or that maintains MarBTN at a level too low to be consistently detected. Further study using experimental challenge of naïve animals will be required to answer this question.

One limitation of the current study is that we observed a high level of mortality in the uninfected clams in the experimental conditions used here, which limited our ability to measure the effect of MarBTN on survival. In natural conditions, soft-shell clam lifespan varies based on latitude and temperature, but they have been reported to live 12–20 years in Massachusetts and Canada, respectively, so the survival observed in the aquarium conditions here is much lower than would be expected in the wild [32]. Future long-term studies of clams should optimize the temperature used to maintain clams to improve the survival of otherwise healthy animals in order to more accurately measure the morbidity and mortality caused by MarBTN.

In addition to observing the different outcomes of MarBTN, this study allows us to measure the timeline of natural disease, showing that there can be a long latent period with low or undetectable levels of cancer in the hemolymph, even in clams that will progress to severe disease and death. Our observation of MarBTN progression in animals that were initially diagnosed as negative using a sensitive qPCR analysis of the hemolymph further demonstrated this latent period. This latent period could be due to quiescent founder cells or to continued cell division that have engrafted into tissues without significant shedding into the hemolymph. The clams observed in this study were infected naturally, so we do not know the exact length of time from initial infection, but as all clams were housed individually without any sharing of the water supply, we know in all cases observed here that infection occurred before Day 0. Because of this, it is also possible that we underestimate the true rate of regression, as clams described here as non-progressors could have previously had higher levels of disease that regressed prior to their collection. Further study of experimental transmission of MarBTN could shed light on the exact length of the incubation time as well as the infectious dose and the initial sites of MarBTN engraftment into a naïve clam.

Our results showing that MarBTN-specific DNA is released into seawater by heavily diseased animals further support the hypothesized water-borne route of cancer cell transmission and provides additional insights into disease transmission dynamics. Despite the correlation with progression, the wide variability in detection is remarkable as well, as it suggests that these cancer cells are likely released in "bursts" rather than continuously released at even levels. Also, the low or undetectable levels produced by clams with lower levels of the cancer suggest that clams early in disease progression are not infectious. It also suggests that "sterilizing immunity" is not necessary to prevent disease transmission in the wild. Evolution of mechanisms that enable clams to keep MarBTN cell replication below the infectious threshold through long-term non-progression or regression could prevent disease spread in the environment.

While it cannot be confirmed that detection by qPCR of MarBTN-specific eDNA in tank water reflects the release of intact, infectious cells, the correlation with disease progression shows that it is consistent with the release of MarBTN cells for heavily diseased animals. We and others have previously shown that BTN cells from soft-shell clams and mussels can survive long-term in seawater conditions similar to those expected in natural environments [19,24], and we have shown that exposure to seawater causes consistent transcriptomic changes in MarBTN cells that may reflect an adaptive response to the water-borne stage of their transmission cycle [20]. Our current results showing correlation between disease progression and eDNA detection provide further support for the hypothesis that MarBTN cells are transmitted through seawater and also validate the use of eDNA assays in ecological studies to detect and track BTNs through eDNA analysis of water around wild populations [26].

In this study, we present data that provide insights into the progression, outcomes, and transmission of BTN in soft-shell clams. This unusual system, in which a cancer lineage itself becomes an infectious disease, has provided a lethal selective pressure on outbred populations of bivalves. This may have led to evolution of unique mechanisms for preventing progression of self-like cancer cells. Identification of the genetic mechanisms by which clams have evolved to prevent cancer progression could initiate a new path for human cancer therapeutics. Additionally, understanding the transmission pathways and the mechanism of evolved resistance to this cancer is critical for understanding the ecological effect of these newly identified infectious cancers and will be important for population management and restoration, both in wild populations and for commercial and subsistence aquaculture.

## Supporting information

**S1 Fig. Agreement between microscopy and qPCR analysis of MarBTN progression.** As shown in Fig 4A–D, dual axis plots compare the amount of MarBTN cells in the hemolymph (red and blue lines, left axis) with quantity of MarBTN-specific eDNA per mL in the tank at the time of collection (black points, right axis). Representative clams are shown for (*A*) progression, (*B*) long-term non-progression, (*C*) regression, and (*D*) negative controls (see S2 Fig for all individual plots). To the right of the qPCR plots are phase-contrast light microscopy images of hemolymph for those same individual representative clams, taken every 4 weeks, after allowing cells to settle in a well of a 96 well plate for 1 hr at 4°C. These images display the variability of morphology of non-neoplastic cells (usually attached to the plate, with extended pseudopodia), and they also show agreement with the quantitative measurements taken at the same time. Images from hemolymph samples that measured >10% MarBTN cells using qPCR are outlined in red. Scale bar in all images is 50 μm.
(TIFF)

**S2 Fig. MarBTN-specific eDNA detection in each clam that was followed for MarBTN in hemolymph and eDNA.** As in the examples shown in Fig 4A–D, eDNA was extracted from tank water collected throughout the study, and MarBTN-specific eDNA was detected using qPCR. Dual axis plots compare the amount of MarBTN cells in the hemolymph (lines, left axis) with quantity of MarBTN-specific eDNA per ml in the tank at the time of collection (black points, right axis). Clams initially diagnosed as naturally low positive (detectible MarBTN < 10% in the hemolymph) have red lines and negative controls clams have blue lines. The animal with clear regression is FFM-21A1. In a second clam (FFM-22H11), the cancer decreased but it did not survive more than 8 weeks after this drop, so under our conservative criteria, we consider it to be progression. All other natural low-positive animals with MarBTN in the hemolymph (n = 8) are animals which progress to death. The slight decrease in MarBTN percent in the hemolymph that is seen in a few cases at the final timepoint before death (FFM-24B11 and FFM-24D2) is likely an artifact due to necrotic release of host DNA into the hemolymph and does not reflect regression of the cancer. Asterisk marks one off-scale eDNA point for FFM-23H7 (1,100 copies/ml at week 14).
(TIFF)

**S3 Fig. Changepoint analysis in comparison of MarBTN-specific eDNA and cancer in hemolymph.** As in Fig 5, for each timepoint for each clam at which both eDNA and hemolymph qPCR results are available, we plot a point comparing the two. The level of MarBTN in the clam is determined by the percent of cells in hemolymph that are MarBTN based on qPCR analysis, and the log transformed copy number of MarBTN-specific eDNA released by clams into their tank water within 24 hrs is plotted on the right panel. The data were also transformed into Bernoulli format (left panel; detectable, 1; undetectable, 0), and 24% was identified as the changepoint (this purple 24% line is plotted on both panels). (TIFF)

**S1 Text. Calculation of equation for estimation of fraction of cancer cells in hemolymph with polyploidy in cancer cells at target site.**
(DOCX)

**S1 Table. Characteristics and outcomes from clams with MarBTN and control healthy clams.**
(XLSX)

**S1 Data. Data for Fig 1.**
(XLSX)

**S2 Data. Data from qPCR analysis of hemolymph from included clams.**
(XLSX)

**S3 Data. Data from qPCR analysis of eDNA and matching hemolymph timepoints.**
(XLSX)

## Acknowledgments

We also thank Brian Beal (University of Maine at Machias) for his advice on maintenance and survival of *Mya arenaria*.

## Author contributions

**Conceptualization:** Rachael M. Giersch, Jordana K. Sevigny, Michael J. Metzger.

**Data curation:** Rachael M. Giersch, Jordana K. Sevigny, Sydney A. Weinandt, Karyn Tindbaek, Marisa A. Yonemitsu, Michael J. Metzger.

**Formal analysis:** Rachael M. Giersch, Jordana K. Sevigny, Sydney A. Weinandt, Carissa Mayo, Karyn Tindbaek, Samuel F. M. Hart, Michael J. Metzger.

**Funding acquisition:** Michael J. Metzger.

**Investigation:** Rachael M. Giersch, Jordana K. Sevigny, Fiona E. S. Garrett, Karyn Tindbaek, Marisa A. Yonemitsu, Samuel FM Hart, Michael J Metzger.

**Methodology:** Rachael M. Giersch, Jordana K. Sevigny, Fiona E. S. Garrett, Samuel F. M. Hart, Michael J. Metzger.

**Project administration:** Michael J. Metzger.

**Supervision:** Michael J. Metzger.

**Validation:** Sydney A. Weinandt.

**Visualization:** Sydney A. Weinandt, Carissa Mayo, Michael J. Metzger.

**Writing – original draft:** Rachael M. Giersch, Jordana K. Sevigny, Sydney A. Weinandt, Michael J. Metzger.

**Writing – review & editing:** Rachael M. Giersch, Jordana K. Sevigny, Sydney A. Weinandt, Carissa Mayo, Fiona E. S. Garrett, Marisa A. Yonemitsu, Samuel F. M. Hart, Michael J. Metzger.

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
