## [Decision Letter · Decision Letter 0]

6 Aug 2025

PPATHOGENS-D-25-01001

Variation in Natural Infection Outcomes and Cancer Cell Release from Soft-Shell Clams (Mya arenaria) with Bivalve Transmissible Neoplasia

PLOS Pathogens

Dear Dr. Metzger,

Thank you for submitting your manuscript to PLOS Pathogens. Once again, I apologize for the long time it took to review the manuscript. After careful consideration, we feel that it has merit but does not fully meet PLOS Pathogens's publication criteria as it currently stands. Therefore, we invite you to submit a revised version of the manuscript that addresses the points raised during the review process.

Please submit your revised manuscript within 60 days Oct 05 2025 11:59PM. If you will need more time than this to complete your revisions, please reply to this message or contact the journal office at plospathogens@plos.org. Please include the following items when submitting your revised manuscript:

We look forward to receiving your revised manuscript.

Kind regards,

Susan R. Ross, PhD

Section Editor

PLOS Pathogens

 Sumita Bhaduri-McIntosh

Editor-in-Chief

PLOS Pathogens

orcid.org/0000-0003-2946-9497

 Michael Malim

Editor-in-Chief

PLOS Pathogens

orcid.org/0000-0002-7699-2064

**Journal Requirements:**

https://journals.plos.org/plospathogens/s/submission-guidelines#loc-parts-of-a-submission

4) We notice that your supplementary Figures, and information are included in the manuscript file. Please remove them and upload them with the file type 'Supporting Information'. Please ensure that each Supporting Information file has a legend listed in the manuscript after the references list.

5) Please ensure that the funders and grant numbers match between the Financial Disclosure field and the Funding Information tab in your submission form. Note that the funders must be provided in the same order in both places as well. Currently, the order of the grants is different in both places.

**Reviewers' Comments:**

Reviewer's Responses to Questions

**Part I - Summary**

Reviewer #1: The manuscript submitted by Giersch, Sevigny et al. is related to the transmissible cancer phenomenon. The study specifically involves bivalve transmissible neoplasia (BTN). BTN is a leukemia-like transmissible cancer, in which the cancerous cells can spread from bivalve to bivalve (clam to clam, in this study). As with other transmissible cancers, the leukemia-like cells are therefore not “self”. Other instances of the phenomenon involve transmission of cells that cause solid tumors in dogs and Tasmanian devils. Thus, BTN is distinct, and can be studied both in the field and in the laboratory.

This report from Giersch et al describes a controlled study of leukemia progression in clams (M. arenaria) naturally infected with MarBTN and provides much needed insights into the variability in uptake and expression. Specifically, the study is significant in that it helps to establish some key parameters of natural infection and disease progression, and does so using naturally infected and uninfected soft-shell clams collected from multiple sites in Maine (which presents its own challenges). From this study, the authors are able to provide insights into variation in progression (and occasional regression), BTN loads in hemolymph, mortality, etc. Among the conclusions, the data suggest that BTN cell numbers have to exceed a certain threshold in the hemolymph before being released into the environment, and the release may occur in “bursts” rather than as a continuous release.

Methodologically, the authors have been rigorous in establishing the experimental and control groups – of some 94 initial clams (47 infected and 47 controls), after strict exclusion the study was initiated with 21 confirmed, naturally low-level infected clams and 39 controls. Three control animals were subsequently found to have low level infection.

The study also includes a brief description of an anecdotal case (lines 219-226).

The introduction also happens to be a strength of this manuscript. Transmissible cancers are unusual as cancers and as pathogens: they are not-quite the same as genetically-induced cancers, but also are distinct from other cellular infectious agents, such as bacteria, parasites and fungi. Thus, this well written, concise introduction is important, as most readers will probably be more familiar with genetically or virally induced neoplasias and/or with more “typical” infectious agents.

Reviewer #2: Re: Review of manuscript # PPATHOGENS-D-25-01001 for PLoS Pathogens “Variation in Natural Infection Outcomes and Cancer Cell Release from Soft-Shell Clams (Mya arenaria) with Bivalve Transmissible Neoplasia” by Rachael M. Giersch*, Jordana K. Sevigny*, Sydney A. Weinandt*, Carissa Mayo, Fiona E. S. Garrett, Karyn Tindbaek, Marisa A. Yonemitsu, Samuel F. M. Hart, Michael J. Metzger.

Summary:

In this manuscript, Giersch and colleagues present a progression-based analysis of a transmissible neoplasia infecting bivalves of the soft-shelled clam species Mya arenaria. Clams were obtained from commercial sources from multiple locations in Maine and status of M. arenaria bivalve transmissible neoplasia (MarBTN) infected/not detectable determined using allele-specific qPCR on DNA extracted from hemolymph. Longitudinal analysis revealed outcomes of infected animals in three main groups including those that progressed to death, long-term non-progressors, and regressors. Analysis of the water for MarBTN by qPCR revealed detectable DNA in levels that correlated with disease progression as inferred via hemolymph. This descriptive study has strengths in supporting the route of transmission via water and variable outcomes including lethality and non-lethality in this species. Major concerns are mostly concerns regarding the overstatement of results. All major and minor concerns of the reviewer are detailed below.

**Part II – Major Issues: Key Experiments Required for Acceptance**

Reviewer #1: No major concerns. The study took on the tough task of evaluating disease parameters in naturally collected (and therefore genetically variable and naturally infected) host clams, which means some parameters could not be known precisely. However, the methods are carefully designed to take these challenges into account, and the caveats are acknowledged in discussion.

Reviewer #2: Major concerns/comments:

A ‘block’ to MarBTN in the presence of this ‘enzootic’ disease of M. arenaria is referenced several times within the text. There is no evidence of the ability of the clams to ‘block cancer’ (e.g., lines 39, 303, 315, 343-345, 359). Rather, it seems more appropriate that such an ability (or the decrease in MarBTN presence using a sensitive allele-specific qPCR approach) is one possible outcome in the presence of MarBTN and should be explored further. This notion is grossly overstated as currently drafted (particularly in the Abstract and lines 343-345 and 359). Please revise for clarity to the audience.

Along the same lines, is anything known concerning the presence and spread (geographical areas, season, general proportion infected) of MarBTN? MarBTN is referred to as ‘enzootic’ twice. As currently presented, the significance to the study of this status is unclear but suggests that MarBTN is constantly present but not epidemic. Is this correct? Does the prevalence of MarBTN in the wild reflect the initial detection of animals used in the study? Given the long term relationship (~200 years), are the progression dynamics altered in any detectable way compared to past studies available for comparison? For example, prior to and now post outbreaks mentioned within the Introduction.

Why is 10% used as a cutoff for detection of MarBTN positive clams?

Imaging of hemolymph is mentioned in the Methods but images are not provided in the manuscript (perhaps this reviewer has overlooked?). Does the imaging of hemolymph reflect the levels of MarBTN detection over time?

What is the typical lifespan (cultured in lab as well as in the wild) of a clam of this species from the 50 mm measurement? Regarding the “limitation of the current study is that we observed a high level of mortality in the uninfected clams in the experimental conditions used here” (lines 323-324), is this consistent with previous work?

Progressors vs. non-progressors vs. regressors. Other apparent limitations of this study are the lack of knowledge of when the clams were initially exposed to MarBTN and status at the start of the experiment (could a clam already have reached a non-progressor status? Have regressed?). Have the authors tried introducing clams without MarBTN detection followed by quarantine (essentially negative controls) to MarBTN positive tanks? (Wouldn’t such an approach confirm the hypothesis that MarBTN is transmitted via the water column?).. Another possible limitation is the collection of the clams, which appears to have been over a timespan ranging from June of 2020 to August 2021 and varies by season. How were the samples selected for inclusion into the study?

Why is 50 mm used as a cutoff? The sizes at measurement ranges from 50 to 73 mm. Did the authors query any correlation of MarBTN detection/outcome with size/(age)?

The ‘anecdotal’ case of the single previously observed regressor maybe requires some clarification. The collection of that clam is prior to those clams collected in the present study (as stated) but appears to coincide with the sample dates from those clams collected in a previous publication, PMID: 35335607. Was the anecdotal clam excluded from a previous study? If so, why? Copper et al are said to have also observed the outcomes of “progression to death at high levels of infection, maintenance of the disease at lower infection levels, and remission from lower levels (Cooper et al., 1982)”. Maybe this reviewer is nitpicking, but it seems that previous study would therefore provide a motivation for the current one, rather than having observed the anecdotal regressor clam. After all, there is one additional ‘regressor’ identified in the present study.

Figure 1. If including these data, axes and units should be revised to reflect those in later figures presenting progression data, e.g., Figure 2.

Use of the term ‘cancer’ is used somewhat generally throughout the text, and sometimes interchangeably with MarBTN. While it is true that MarBTN is a clonal leukemia-like cancer of the bivalve M. arenaria, the general use for the term ‘cancer’ may at times be misleading to the audience. For example, the study is centered on MarBTN detection via a sensitive qPCR assay. Please revise the text to reflect MarBTN detection and measurements, rather than ‘cancer’ progression dynamics. Is ‘progression’ only indicated by an increase or decrease in MarBTN levels? Are there any other indicators or markers thereof?

Line 36 and throughout the text: “…cancer cell release…”. To this reviewer, the results measure DNA using the allele-specific primers in qPCR, but do not demonstrate the presence of ‘cancer cells’ explicitly. Please revise the wording to reflect what is being shown.

Line 104: the “release of BTN” sounds rather directed as currently stated; suggest “presence of BTN”.

Line 104: “at different stages of the disease”… By ‘stages’, do the authors just mean the levels of MarBTN as interpreted via qPCR? A statement here reflecting that point near lines 278-279 concerning observations indicating two stages could be of help to the audience and bolster the significance of the results overall.

Line 256: The header of this section should be edited to reflect MarBTN levels as interpreted through the qPCR levels rather than ‘Release of cancer cells into water’. In the same section (and as raised by this reviewer elsewhere here), statements should be revised to reflect MarBTN levels rather than high/low amounts of cancer.

Line 350: Suggest removing the word “clear”.

**Part III – Minor Issues: Editorial and Data Presentation Modifications**

Reviewer #1: Line 151-152 “Water samples from the tanks of animals FFM-18E12, FFM-16E1, and FFM-15A12 received a slightly modified protocol and were incubated for 30 min without shaking”. Could the authors include a sentence as to why? Would the reason have any effect on the outcome or interpretation?

245-249 The authors suggest that the three controls that turned positive reflect a long latent period or period in which infection is localized and not detectable in hemolymph. I wonder if “latent” is the appropriate term here? Does latent include the possibilities of quiescent founder BTN cells and/or ongoing or significant cell division but in a localized environment?

The into and discussion didn’t mention the possibility that variation in the MarBTN population could contribute to differences in progression and outcome. Since the clams were collected from the environment and from different sites, is it possible that another confounding/contributing factor could be variation in the MarBTN cells from clam to clam? This is not a major point, but if this is a possibility or if there is any knowledge related to MarBTN variability, it should be briefly discussed.

Reviewer #2: Minor Concerns/comments:

Line 40: “This also”… this what?

Line 46: “there are three types of transmissible cancers that occur in nature” suggest editing to read “at least three types of transmissible cancers have been observed” or similar.

Line 58-60: Is there a missing reference for this statement?

Line135: Suggest removing the statement “After data collection, clams were excluded for several reasons”.

Line 151: Typo, ‘Water’ should be lower case.

Lines 163-164: The primers are previously published and that citation can be provided.

Line 168: Suggest editing to remove the last portion of the statement corresponding to “and has not been detected in any healthy clam”.

Line 245-247: Suggest removing “healthy” to reflect ‘negative controls’ alone… The statement “MarBTN in these three animals shows that there is a significant latent period in natural disease”. Is this true? Do the findings rather suggest that a long latency period may sometimes be present?

Paragraph near Line 251: The statements should be edited to reflect appropriate tense.

PLOS authors have the option to publish the peer review history of their article (what does this mean?). If published, this will include your full peer review and any attached files.

Reviewer #1: No

Reviewer #2: No

**Figure resubmission:**
---

## [Editor Report · Decision Letter 1]

16 Sep 2025

Dear Dr. Metzger,

We are pleased to inform you that your manuscript 'Variation in Natural Infection Outcomes and Cancer Cell Release from Soft-Shell Clams (Mya arenaria) with Bivalve Transmissible Neoplasia' has been provisionally accepted for publication in PLOS Pathogens.

Best regards,

Susan R. Ross, PhD

Section Editor

PLOS Pathogens

Susan Ross

Section Editor

PLOS Pathogens

Sumita Bhaduri-McIntosh

Editor-in-Chief

PLOS Pathogens

orcid.org/0000-0003-2946-9497

Michael Malim

Editor-in-Chief

PLOS Pathogens

orcid.org/0000-0002-7699-2064
---

## [Editor Report · Acceptance letter]

Dear Dr. Metzger,

We are delighted to inform you that your manuscript, "Variation in Natural Infection Outcomes and Cancer Cell Release from Soft-Shell Clams (Mya arenaria) with Bivalve Transmissible Neoplasia," has been formally accepted for publication in PLOS Pathogens.

Best regards,

Sumita Bhaduri-McIntosh

Editor-in-Chief

PLOS Pathogens

orcid.org/0000-0003-2946-9497

Michael Malim

Editor-in-Chief

PLOS Pathogens

orcid.org/0000-0002-7699-2064